# The Moderating Role of the *FKBP5* Gene Polymorphisms in the Relationship between Attachment Style, Perceived Stress and Psychotic-like Experiences in Non-Clinical Young Adults

**DOI:** 10.3390/jcm11061614

**Published:** 2022-03-15

**Authors:** Filip Stramecki, Błażej Misiak, Łukasz Gawęda, Katarzyna Prochwicz, Joanna Kłosowska, Jerzy Samochowiec, Agnieszka Samochowiec, Edyta Pawlak, Elżbieta Szmida, Paweł Skiba, Andrzej Cechnicki, Dorota Frydecka

**Affiliations:** 1Department of Psychiatry, Wroclaw Medical University, Pasteur Street 10, 50-367 Wroclaw, Poland; fstramecki@gmail.com; 2Department of Psychiatry, Division of Consultation Psychiatry and Neuroscience, Wroclaw Medical University, Pasteur Street 10, 50-367 Wroclaw, Poland; blazej.misiak@umed.wroc.pl; 3Experimental Psychopathology Lab, Institute of Psychology, Polish Academy of Sciences, Jaracza Street 1, 00-378 Warsaw, Poland; l.gaweda@psych.pan.pl; 4Institute of Psychology, Jagiellonian University, Ingardena 6 Street, 30-060 Krakow, Poland; katarzyna.prochwicz@uj.edu.pl (K.P.); joanna.klosowska@uj.edu.pl (J.K.); 5Department of Psychiatry, Pomeranian Medical University, Broniewskiego 26 Street, 71-457 Szczecin, Poland; samoj@pum.edu.pl; 6Institute of Psychology, Department of Clinical Psychology, University of Szczecin, 71-017 Szczecin, Poland; agnieszkasamochowiec@gmail.com; 7Department of Experimental Therapy, Institute of Immunology and Experimental Therapy, Polish Academy of Sciences, Weigla Street 12, 53-114 Wroclaw, Poland; edyta.pawlak@hirszfeld.pl; 8Department of Genetics, Wroclaw Medical University, Marcinkowskiego 1 Street, 50-368 Wroclaw, Poland; e.szmida@gmail.com (E.S.); pawel.skiba@umed.wroc.pl (P.S.); 9Department of Community Psychiatry, Medical College Jagiellonian University, Sikorskiego Place 2, 31-115 Krakow, Poland; acechnicki@interia.pl

**Keywords:** psychosis, genetics, HPA-axis, stress, attachment, FKBP5

## Abstract

Numerous studies have reported that stressful life experiences increase the risk of psychosis and psychotic-like experiences (PLEs). Common variations of the *FKBP5* gene have been reported to impact the risk of psychosis by moderating the effects of environmental exposures. Moreover, anxious and avoidant attachment styles have been shown to increase both the level of perceived stress and the risk for psychosis development. In the present cross-sectional study, we aimed to investigate whether variants of the *FKBP5* gene moderate the effects of attachment styles and the level of perceived stress on the development of PLEs. A total of 535 non-clinical undergraduates were genotyped for six *FKBP5* single nucleotide polymorphisms (SNPs) (rs3800373, rs9470080, rs4713902, rs737054, rs1360780 and rs9296158). The Psychosis Attachment Measure (PAM), the Perceived Stress Scale-10 (PSS-10) and the Prodromal Questionnaire 16 (PQ-16) were administered to assess attachment styles, the level of perceived stress and PLEs, respectively. Anxious attachment style, lower levels of perceived self-efficacy and higher levels of perceived helplessness were associated with a significantly higher number of PLEs. The main effects of attachment style on the severity of PLEs were significant in models testing for the associations with perceived self-efficacy and three FKBP5 SNPs (rs1360780, rs9296158 and rs9470080). The main effect of rs38003733 on the number of PLEs was observed, with GG homozygotes reporting a significantly higher number of PLEs in comparison to T allele carriers. In individuals with dominant anxious attachment style, there was a significant effect of the interaction between the *FKBP5* rs4713902 SNP and self-efficacy on the severity of PLEs. Among rs4713902 TT homozygotes, a low level of perceived self-efficacy was associated with higher severity of PLEs. In subjects with non-dominant anxious attachment, a low level of perceived self-efficacy was associated with a higher number of PLEs, regardless of the genotype. Our results indicate that the *FKBP5* gene might moderate the relationship between attachment, perceived stress and PLEs.

## 1. Introduction

In past decades, numerous studies have focused on risk factors for the development of psychosis and have demonstrated the importance of genetic and environmental factors [1,2,3]. According to the continuum model [4], psychotic-like experiences (PLEs) are being described as subclinical psychotic symptoms which can be present in non-clinical populations with the prevalence rate estimated at 7.2% in the general population [5].

Several models have been proposed so far in the development of PLEs, showing the association of genetic background [6,7], cannabis use [8], cognitive biases [9,10], self-disturbances [11], insecure attachment style [12] and early traumatic adversities [13] with the higher risk of PLEs [14]. Moreover, it has been reported that childhood trauma and PLEs are associated with increased suicidal risk in young adults [15].

Elevated levels of stress and increased stress sensitivity have been found to serve as important psychosocial risk factors for psychosis development. Exposure to stressful life experiences in childhood has been reported more frequently in individuals with PLEs than in the general population [16]. A greater level of perceived stress has been correlated with a higher frequency of PLEs, and this relationship has been found to be mediated by maladaptive coping strategies [17]. It has been shown that using maladaptive coping strategies makes individuals more prone to evaluate a neutral situation as stressful and increases the level of perceived stress [17]. On the other hand, the level of perceived stress has been found to be associated with a greater likelihood to report PLEs [18]. Moreover, several studies have shown that individuals with anxious attachment styles tend to use more ineffective coping strategies, and are more likely to perceive higher levels of stress [19,20,21].

Among psychosocial factors involved in psychosis development, insecure attachment styles have been shown to increase the risk for clinical psychosis [22,23,24,25,26] as well as the risk of PLEs [23,24,27]. Attachment theory proposed by Bowlby describes the impact of early interactions of a newborn with their primary caregivers on an individual’s psychological functioning in later life and defines attachment style as a mental representation of the self in relation to others [28]. Insecure attachment—that includes anxious and avoidant attachment style—is characterized by the failure to relieve distress by proximity seeking and triggers the use of secondary attachment strategies [29]. Attachment anxiety involves a strong need for closeness, worries about relationships, and fear of being rejected, leading to the development of hyperactive strategies for regulating distress, such as intense attempts to maintain proximity and support in the relation to others [29,30]. In turn, the avoidant attachment style is associated with compulsive self-reliance, and preference for emotional distance from others, need for being independent as well as difficulties with close relations and intimacy [29,30].

Since the hypothalamic–pituitary–adrenal (HPA) axis is the main endogenous system involved in the stress response, it is likely to play a key role in mediating the effect of attachment style on the human stress response [31]. Stress directly activates the HPA axis activity and triggers several other hormonal responses. Exposure to early-life stress, entailing the dysregulation of HPA axis activity has an important influence on stress response later in life, which corresponds to an increased level of diurnal cortisol [32]. Elevated cortisol levels have been observed both in patients with first-episode psychosis [33,34] and in non-clinical at-risk subjects [35] when compared to heathy controls. Chronically elevated cortisol levels resulting from adverse environments and poor parenting can affect the functioning of the HPA axis and has been associated with insecure attachment development [31,36].

It has been found that the HPA axis activity is modulated by a number of *FKBP5* gene variants. The *FKBP5* gene encodes the FK506 binding protein 51 (FKBP5), which is a co-chaperone of the glucocorticoid receptor (GR) [37]. The FKBP5 binds to GR and diminishes its affinity to cortisol, which modulates individual stress response [38]. Several single nucleotide polymorphisms (SNPs) in the *FKBP5* gene have been found to affect stress-related psychiatric conditions [39] mostly by moderating the effect of traumatic experiences on the severity of the psychiatric symptomatology [40]. SNPs rs9296158 and rs4713016 were found to increase the risk for psychosis development by affecting cortisol levels in response to trauma and rs9296158 was associated with the development of more severe psychotic symptoms [41]. Our recent study on a non-clinical sample supports the moderating role of SNPs rs9296158, rs1360780 and rs737054 on PLEs severity in response to early life trauma [42].

Although there are reports showing relationships between the *FKBP5* gene polymorphisms, attachment styles, perceived stress, and the severity of PLEs, to our knowledge, there has been no study analyzing all these variables interacting together in one comprehensive model. In accordance with previous studies, we hypothesized that the effect of attachment style and level of perceived stress on PLEs development might be moderated by specific SNPs of the *FKBP5* gene in a non-clinical sample of young adults.

## 2. Materials and Methods

### 2.1. Participants

A total of 535 participants aged 23.4 ± 3.4 years (range: 18–30 years) were recruited among university students of various faculties from three large Polish cities (Kraków, Wrocław and Szczecin) in the years 2017–2019. All participants represented Caucasian ethnicity and were not related to each other. The history of clinical diagnosis and frequency of substance use were provided with a self-report questionnaire designed for this study. No participant reported a history of psychosis spectrum disorder. Frequent substance use was defined as the use of any psychoactive substance, including alcohol, more than once a week. The study has a cross-sectional design. The Ethics Committee at Wroclaw Medical University in Poland approved the study (approval number: 254/2018; issued on 19th July 2018). All participants gave written informed consent for participation in the study.

### 2.2. Measures

#### 2.2.1. The Prodromal Questionnaire 16 (PQ-16)

The 16-item PQ-16 is a self-report screening tool for at-risk mental states and the presence of PLEs with sensitivity and specificity estimated at 87% [43]. It consists of nine items assessing perceptual alterations, five items to investigate delusional ideation, paranoia and unusual thought, and two items to screen for attenuated negative symptoms. In our study, we used the Polish version of the questionnaire that was prepared using a back-translation procedure and was used in our several previous studies [6,12,18]. The original PQ-16 consists of two scales, where the first one records whether PLEs are “present” or “non-present”, and the second one assesses the level of distress associated with PLEs on a four-point Likert-like scale. In the present study, we focused on the number of PLEs being reported by each participant. Therefore, the total score on the PQ-16 was calculated by adding up the agreed items. The Cronbach’s alpha of the PQ-16 was 0.75 in our sample, indicating acceptable internal consistency.

#### 2.2.2. Psychosis Attachment Measure (PAM)

The Psychosis Attachment Measure (PAM) is a 16-item self-report questionnaire [44]. It records two different attachment styles: anxious and avoidant. There are eight items measuring attachment anxiety and eight items recording attachment avoidance. Participants are asked how they relate to the key people from their life on a four-point Likert. Internal consistency was acceptable (Cronbach’s alpha = 0.80) for the anxious attachment subscale and good (Cronbach’s alpha = 0.82) for the avoidance attachment subscale. In the present study, we used the Polish version of the questionnaire [45]. Based on the PAM scoring, we divided the respondents into those with dominant attachment anxiety and others.

#### 2.2.3. Perceived Stress Scale-10 (PSS-10)

The level of perceived stress was assessed using the Perceived Stress Scale-10 (PSS-10) [46]. The PSS-10 is a 10-item self-report questionnaire designed to measure the degree that respondents find their life unpredictable, uncontrollable, and overwhelming in the preceding month. It consists of 10 items rating the frequency with which participants experience certain situations. The items are scored on a five-point Likert-like scale with responses ranging from “never” to “very often”. The PSS-10 consists of two subscales measuring the level of self-efficacy (PSE) and helplessness (PHS). Internal consistency of the PSS-10 in our sample was acceptable (the Cronbach’s alpha = 0.74). We used the Polish version of the PSS-10 and measured the intensity of perceived stress related to the current life situation, specifically 30 days prior to the assessment. Participants were divided into those with the PSS-10 score above and below the mean value.

### 2.3. Genotyping

Based on the functional impact on the *FKBP5* gene and HPA-axis activity, we selected six *SNPs* (rs3800373, rs9470080, rs4713902, rs737054, rs1360780 and rs9296158). DNA samples were collected by swabs from the inner cheeks. We confirmed the accuracy of genotypes by performing duplicate genotyping for 25% of randomly selected samples. Subjects performing the genotyping were blinded to the ID of participants and the data collected using specific questionaries. The details of the genetic analysis were described in our previous study [42].

### 2.4. Statistical Analysis

The χ^2^ test was used to assess the agreement of genotype distribution with the Hardy–Weinberg equilibrium (HWE). Correlations between continuous variables were tested using the Spearman rank correlation coefficients. The Mann–Whitney U test was employed to perform bivariate comparisons. Due to potential effects of age, sex, a history of clinical diagnosis and substance use, the analysis of co-variance (ANCOVA) was performed. Before running the ANCOVA, participants were divided into individuals with predominant anxious attachment (scores of anxious attachment higher than those for avoidant attachment) and those employing other attachment styles (scores of avoidant attachment higher than or equal to those for anxious attachment). Similarly, the PSS scores of both subscales (perceived helplessness and perceived self-efficacy) were dichotomized based on the mean values (i.e., participants were divided into those with the PSS scores above and below the mean value). In the first step, the association between exposure (low vs. high PSS score) and outcome (the PQ-16 score) was tested. Next, covariates (age, gender, a history of clinical diagnosis and frequent substance use) were added to the model. Finally, the following moderators were added: (1) main effects (the *FKBP5* gene polymorphisms and predominant attachment style); (2) two-way interactions (the *FKBP5* polymorphism × the PSS score; the *FKBP5* polymorphism × predominant anxious attachment and the PSS score × predominant anxious attachment) and (3) the three-way interaction (the *FKBP5* polymorphism × the PSS score × predominant anxious attachment). In case of significant interactions, the Games–Howell test was applied to perform post-hoc comparisons. The level of significance was set at *p* < 0.05. All analyses were carried out using the Statistical Package for Social Sciences, version 20 (SPSS Inc., Chicago, IL, USA).

## 3. Results

The general characteristics of all participants are presented in Table 1. Out of 535 participants enrolled in the present study, 460 individuals provided complete data on the occurrence of PLEs, attachment styles and the level of perceived stress (86.2%). Sufficient quality and quantity of DNA was obtained for 441–450 participants (82.4–83.9%) depending on the specific SNPs. Predominant anxious attachment style was reported by 186 individuals (40.4%). The PSS-10 scores of perceived self-efficacy and helplessness were 10 ± 2.90 and 12 ± 5.19, respectively. Clinical diagnosis, including mood or anxiety disorders, was reported by 8.2% of the participants. None of these respondents reported to be diagnosed with a psychotic spectrum disorder. Frequent use of substances was reported by 21.1% of all participants. 

Direct effects of exposure (categories of perceived stress) are shown in Table 2. Participants with lower levels of perceived self-efficacy had a significantly higher number of PLEs (4.1 ± 2.9 vs. 2.9 ± 2.6), even after controlling for age, gender, a history of clinical diagnosis and frequent substance use. Similarly, higher levels of perceived helplessness were associated with a significantly higher number of PLEs (4.3 ± 2.9 vs. 2.8 ± 2.5), even after co-varying for age, gender, a history of clinical diagnosis and frequent substance use. Individuals with a predominant anxious attachment style had a significantly higher number of PLEs (4.0 ± 2.9 vs. 3.2 ± 2.7, *p* = 0.001).

Table 3 presents the results from the analysis of the relationships between the SNPs of the *FKBP5* gene, attachment styles and perceived stress. There were significant effects of the two-way interaction (*FKBP5* rs4713902 polymorphism × perceived self-efficacy) and the three-way interaction (*FKBP5* rs4713902 polymorphism × perceived self-efficacy × attachment) on the number of PLEs. The main effects of age, perceived stress and clinical diagnosis were significant in all models. The main effects of predominant attachment were significant only in three models testing for the associations with perceived self-efficacy and three *FKBP5* polymorphisms (rs1360780, rs9296158 and rs9470080). A significant main effect of the rs3800373 polymorphism on the number of PLEs was observed. Specifically, the rs3800373 GG homozygotes reported a significantly higher number of PLEs compared to the rs3800373 T allele carriers (5.1 ± 3.0 vs. 3.4 ± 2.7, *p* < 0.001).

Post-hoc comparisons are reported in Figure 1. Among the rs4713902 TT homozygotes, a low level of perceived self-efficacy was associated with a significantly higher number of PLEs. The rs4713902 TT homozygotes with a low level of perceived self-efficacy had a significantly higher number of PLEs compared to the rs4713902 C allele carriers with high and low levels of perceived self-efficacy. Further stratification of the sample demonstrated that a low level of perceived self-efficacy is associated with a significantly higher number of PLEs only in subjects with the rs4713902 TT genotype and predominant anxious attachment style. Among individuals with non-dominant anxious attachment style, a low level of perceived self-efficacy was associated with a significantly higher number of PLEs, regardless of the rs4713902 genotype.

## 4. Discussion

We found that the predominant anxious attachment style is associated with a significantly higher number of PLEs. The development of an anxious attachment style is an effect of unprotective parenting corresponding to early life environmental stressors. This may contribute to neurodevelopmental alterations which lead to a higher likelihood of PLEs development [47]. This stays in line with previous studies showing that anxious attachment style increases the risk for PLEs development in individuals exposed to poor parenting [23,24]. Associations between attachment style and PLEs development are supported by studies on patients with schizophrenia-spectrum disorders showing that insecure attachment style mediates the influence of traumatic experiences on the severity of psychotic symptoms in schizophrenia-spectrum disorders [22,26,48].

It is worth noting that we observed a specific influence of attachment anxiety rather than attachment avoidance on PLEs development. This has been previously proposed as an explanation of why symptoms remain subthreshold with individuals with anxious attachment and do not lead to full blown psychosis. Avoidant behaviors observed in negative psychotic symptoms predispose to social withdrawal which leads to growing relational impairments while people with anxious attachment remain involved in social life which may protect them to develop clinical psychosis [49].

Moreover, we observed that participants with lower levels of perceived self-efficacy and higher levels of perceived helplessness reported a higher number of PLEs which supports previous reports on the positive correlation between perceived stress level and PLEs development [17,18,50]. This may indicate that increased stress level makes individuals more vulnerable to developing PLEs. However, it might also signify that PLEs themselves are responsible for elevated levels of perceived stress.

We observed that rs3800373 GG homozygotes are more likely to report a higher number of PLEs in comparison to the rs3800373 T allele carriers. This stays in line with a previous study showing that carrying the rs3800373 G allele of the *FKBP5* gene is associated with a higher risk for schizophrenia development [51]. Moreover, the C allele of rs3800373 has been found to promote higher attachment insecurity in response to parenting insensitivity [52]. These results suggest that *FKBP5* rs3800373 SNP plays an important role in the phenomenon of the psychosis continuum. Considering that the *FKBP5* gene moderates the brain response and HPA-axis reactivity to stress, it seems reasonable that it is also responsible for the variable susceptibility to PLEs development. However, the body of research on *FKBP5* polymorphism is too small to draw conclusions about the exact mechanisms of specific SNPs on PLEs.

We have found that the main effects of attachment style on the severity of PLEs were significant in models testing for the associations with perceived self-efficacy and three *FKBP5* SNPs (rs1360780, rs9296158 and rs9470080). The results of the present study support previous reports showing the relationship between *FKBP5* gene polymorphisms and attachment [52,53,54,55]. Only a few studies assessed *FKBP5* gene polymorphism in the context of attachment. It has been demonstrated that insecure attachment is associated with greater cortisol reactivity levels in T allele carriers of rs1360780 *FKBP5* gene polymorphism [53,54,55]. Accordingly, individuals carrying the T allele of rs1360780 were more likely to develop an insecure attachment style, and thus experience more difficulties in coping when compared to the CC homozygotes [54]. This may suggest that early life adversities like poor parenting and an unprotective environment are responsible for altered neurodevelopment and dysregulation of the stress-response system which results in higher proneness to PLEs and impaired stress perception in adult life.

In individuals with dominant anxious attachment style, there was a significant effect of the interaction between the *FKBP5* rs4713902 SNP and self-efficacy on the severity of PLEs. Among rs4713902 TT homozygotes, a low level of perceived self-efficacy was associated with higher severity of PLEs. In subjects with non-dominant anxious attachment, a low level of perceived self-efficacy was associated with a higher number of PLEs, regardless of the genotype. Both attachment anxiety and a high level of perceived stress have been found to be associated with greater development of PLEs [56,57]. Therefore, considering that the *FKBP5* gene has been implicated in the development and severity of psychosis and PLEs [41,42], it may suggest that anxious attachment style is associated with different neurobiological mechanisms compared to other insecure attachment styles i.e., avoidant attachment. Although we observed significant relationships between attachment, level of perceived stress and PLEs it is impossible to establish the exact causality of PLEs development. Disturbed attachment results from harmful and non-supportive environment considered as early life adversity and may itself predispose to greater stress perception, maladaptive coping strategies and leads to higher psychosis proneness. Nevertheless, our results indicate that *FKBP5* gene polymorphisms are responsible mostly for different stress responses and HPA-axis reactivity plays a moderating role in these relationships. Therefore, it can be hypothesized that carrying the T allele of SNP rs4713902 of the FKBP5 gene increases the risk for attachment anxiety which may further lead to both increased level of perceived stress and greater risk for PLEs.

However, the findings of the present study should be interpreted in the light of several limitations. First, we assessed only six SNPs. Hence, this may not fully represent the moderating effect of the *FKBP5* gene on assessed variables. Second, our study had a relatively limited sample size. Furthermore, it is important to note that the assessment of the variables was based on self-report questionnaires. This could cause recall bias which might be characterized by overestimation of PLEs reported by respondents [58]. Moreover, we did not exclude participants with a history of clinical diagnosis. Taking into consideration that the proportion of variance in the level of PLEs was relatively low, this may suggest that other factors might be associated with PLEs. However, it is important to highlight that none of the participants reported to be diagnosed with psychosis spectrum disorders. Although the study was based on a non-clinical sample which may increase generalizability to the general population, it might also limit the translation of findings to clinical samples. However, the generalizability of the present results may also be limited by a slightly higher number of female participants. Finally, a cross-sectional study design does not allow to draw conclusions on causal effects. Further research should investigate both non-clinical and clinical samples to identify whether the moderating effect of variants in the *FKBP5* gene on presented associations is relevant for the whole psychosis continuum. Likewise, future studies may assess a higher number of SNPs in larger samples to enable a relevant generalization of findings. Although we did not observe direct relationships between specific insecure attachment styles and negative and positive PLEs it might be worth considering examining these relationships in further research.

## 5. Conclusions

Despite several limitations, our results provide a novel contribution by showing that the *FKBP5* gene plays a moderating role in the relationships between perceived stress and attachment style in the context of PLEs, increasing the risk of their development in TT homozygotes of single nucleotide polymorphism rs4713902 with both low-level self-efficacy and anxious attachment style when compared to C allele carriers. The GG homozygotes of SNP rs3800373 are more likely to report PLEs than T allele carriers. Non-dominant anxious attachment style and a low level of perceived self-efficacy were associated with a higher number of reported PLEs. These results may indicate the importance of exploring the field of the *FKBP5* gene role in the development of PLEs to enable the design of novel therapeutic directions which may focus on psychosis prevention. The present results suggest that simple relations between attachment style and perceived stress level may not be sufficient to comprehend its impact on the development of subthreshold psychotic symptoms. The findings imply that candidate genes involved in stress-response may play a pivotal role in moderating these associations. Future studies on larger samples and based on extensive genetic assessment should consider replication of the findings and include also other domains of the psychosis continuum.

## Figures and Tables

**Figure 1 jcm-11-01614-f001:**
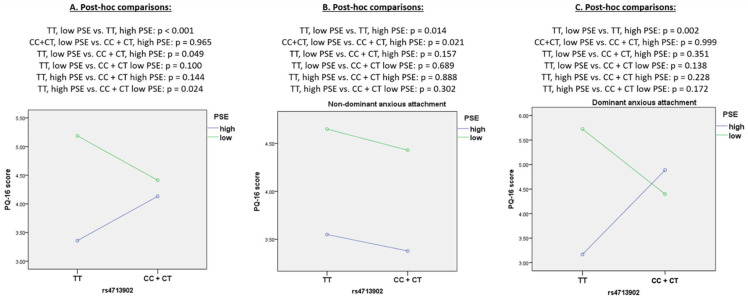
Post-hoc comparisons of interactions between *FKBP5* rs4713903 polymorphism total PQ-16 score and attachment style. Abbreviations: PSE—perceived self-efficacy, PQ-16—Prodromal Questionnaire 16.

**Table 1 jcm-11-01614-t001:** General characteristics of the sample.

	Mean ± SD or n (%)
Age, years	23.4 ± 3.0
Gender, M/F	133/327 (40.7/59.3)
Clinical diagnosis	38 (8.2)
Anxious attachment	1.22 ± 0.65
Avoidant attachment	1.21 ± 0.65
Predominant anxious attachment	186 (40.4)
Perceived helplessness	12 ± 5.19
Perceived self-efficacy	10 ± 2.90
Frequent use of substances (>once per week)	97 (21.1)
PQ-16	4.1 ± 4.6
rs1360780	450
CC	260 (58.56)
CT	159 (34.46)
TT	31 (6.98)
rs9296158	444
AA	26 (5.84)
AG	159 (35.73)
GG	260 (58.43)
rs3800373	443
GG	37 (8.35)
TG	144 (32.51)
TT	262 (59.14)
rs9470080	443
CC	245 (55.30)
CT	151 (34.09)
TT	47 (10.61)
rs4713902	441
CC	50 (11.34)
CT	154 (34.92)
TT	237 (53.74)
rs737054	449
CC	224 (49.89)
CT	182 (40.53)
TT	43 (9.58)

**Table 2 jcm-11-01614-t002:** Effects of perceived stress on the PQ-16 score after adjustment for general characteristics of the sample.

Model	Effect	Perceived Helplessness	Perceived Self-Efficacy
Model 1 (exposure)	Perceived stress	F = 43.90, ***p*** **< 0.001**	F = 25.90, ***p*** **< 0.001**
	R^2^	0.088	0.054
Model 2 (exposure and covariates)	Perceived stress	F = 24.62, ***p*** **< 0.001**	F = 12.90, ***p*** **< 0.001**
	Age	F = 39.21, ***p*** **< 0.001**	F = 42.35, ***p*** **< 0.001**
	Sex	F = 0.68, *p* = 0.412	F = 0.60, *p* = 0.441
	Clinical diagnosis	F = 4.63, *p* = 0.032	F = 5.29, *p* = 0.022
	Frequent substance use	F = 2.09, *p* = 0.149	F = 2.92, *p* = 0.09
	R^2^	0.178	0.155

**Table 3 jcm-11-01614-t003:** Interactions between the *FKBP5* gene polymorphisms, perceived stress and attachment style (effects of exposure, covariates and moderators).

Stress Category	Effect	rs1360780	rs9296158	rs3800373	rs9470080	rs4713902	rs737054
Perceived helplessness	Age	F = 34.11; ***p*** **< 0.001**	F = 33.28; ***p*** **< 0.001**	F = 29.09; ***p*** **< 0.001**	F = 30.96; ***p*** **< 0.001**	F = 29.61; ***p*** **< 0.001**	F = 29.83; ***p*** **< 0.001**
	Gender	F < 0.01; *p* = 0.940	F < 0.01; *p* = 0.956	F = 0.02; *p* = 0.889	F = 0.01; *p* = 0.922	F = 0.01; *p* = 0.915	F = 0.03; *p* = 0.854
	Clinical diagnosis	F = 4.51; ***p*** **= 0.034**	F = 4.32; ***p*** **= 0.038**	F = 5.76; ***p*** **= 0.017**	F = 4.64; ***p*** **= 0.032**	F = 4.88; ***p*** **= 0.028**	F = 4.19; ***p*** **= 0.041**
	Frequent substance use	F = 2.85; *p* = 0.092	F = 2.90; *p* = 0.089	F = 2.21; *p* = 0.138	F = 2.05; *p* = 0.153	F = 1.28; *p* = 0.259	F = 1.93; *p* = 0.165
	Perceived helplessness	F = 19.68; ***p*** **< 0.001**	F = 21.62; ***p*** **< 0.001**	F = 7.00; ***p*** **= 0.009**	F = 20.53; ***p*** **< 0.001**	F = 20.70; ***p*** **< 0.001**	F = 23.14; ***p*** **< 0.001**
	Attachment	F = 3.23; *p* = 0.073	F = 3.25; *p* = 0.072	F = 0.62; *p* = 0.433	F = 3.62; *p* = 0.058	F = 2.79; *p* = 0.096	F = 2.85; *p*= 0.092
	*FKBP5*	F = 1.11; *p* = 0.294	F = 2.36; *p* = 0.126	F = 8.82; ***p*** **= 0.003**	F = 0.81; *p* = 0.370	F = 0.10; *p* = 0.748	F = 0.16; *p* = 0.687
	Perceived helplessness × attachment	F = 0.15; *p* = 0.696	F = 0.06; *p* = 0.811	F = 0.39; *p* = 0.534	F = 0.17; *p* = 0.684	F = 0.80; *p* = 0.779	F = 0.08; *p* = 0.776
	*FKBP5* × attachment	F = 0.23; *p* = 0.629	F = 0.06; *p* = 0.808	F = 0.03; *p* = 0.875	F = 0.12; *p* = 0.732	F = 0.25; *p* = 0.615	F = 0.52; *p* = 0.473
	*FKBP5* × perceived helplessness	F < 0.01; *p* = 0.996	F = 0.06; *p* = 0.813	F < 0.01; *p* = 0.989	F = 0.23; *p* = 0.634	F = 3.49; *p* = 0.062	F = 1.66; *p* = 0.199
	*FKBP5* × perceived helplessness × attachment	F = 0.57; *p* = 0.452	F = 1.02; *p* = 0.314	F = 1.34; *p* = 0.238	F = 1.32; *p* = 0.251	F = 1.64; *p* = 0.201	F = 1.80; *p* = 0.180
	R^2^	0.189	0.192	0.207	0.182	0.184	0.192
Perceived self-efficacy	Age	F = 36.37; ***p*** **< 0.001**	F = 35.61, ***p*** **< 0.001**	F = 31.18; ***p*** **< 0.001**	F = 33.53; ***p*** **< 0.001**	F = 31.60; ***p*** **< 0.001**	F = 33.00; ***p*** **< 0.001**
	Gender	F < 0.01; *p* = 0.991	F < 0.01; *p* = 0.985	F < 0.01; *p* =0.990	F < 0.01; *p* = 0.937	F = 0.00; *p* = 0.979	F = 0.02; *p* = 0.901
	Clinical diagnosis	F = 5.69; ***p*** **= 0.018**	F = 5.46; ***p*** **= 0.020**	F = 6.11; ***p*** **= 0.014**	F = 5.70; ***p*** **= 0.017**	F = 6.33; ***p*** **= 0.012**	F = 5.32; ***p*** **= 0.022**
	Frequent substance use	F = 3.71; *p* = 0.055	F = 3.59; *p* = 0.059	F = 3.28; *p* = 0.071	F = 2.75; *p* = 0.098	F = 1.97; *p* = 0.161	F = 2.84; *p* = 0.093
	Perceived self-efficacy	F = 10.72; ***p*** **= 0.001**	F = 11.46; ***p*** **= 0.001**	F = 4.60; ***p*** **= 0.033**	F = 11.00; ***p*** **= 0.001**	F = 12.00; ***p*** **= 0.001**	F = 11.04; ***p*** **= 0.001**
	Attachment	F = 3.96; ***p*** **= 0.047**	F = 4.46; ***p*** **= 0.035**	F = 0.52; *p* = 0.474	F = 4.39; ***p*** **= 0.037**	F = 3.02; *p* = 0.083	F = 3.71; *p* = 0.055
	*FKBP5*	F = 0.73; *p* = 0.394	F = 1.74; *p* = 0.188	F = 8.78; ***p*** **= 0.003**	F = 0.24; *p* = 0.626	F < 0.01; *p* = 1.000	F = 0.18; *p* = 0.668
	Perceived self-efficacy × attachment	F = 0.06; *p* = 0.803	F = 0.01; *p* = 0.920	F = 0.02; *p* = 0.884	F = 0,02; *p* = 0.882	F = 0.01; *p* = 0.945	F = 0.17; *p* = 0.678
	*FKBP5* × attachment	F = 0.01; *p* = 0.924	F = 0.19; *p* = 0.891	F = 0.14; *p* = 0.714	F = 0.01; *p* = 0.917	F = 0.42; *p* = 0.519	F = 0.40; *p* = 0.528
	*FKBP5* × perceived self-efficacy	F = 2.53; *p* = 0.113	F = 1.93; *p* = 0.166	F = 0.09; *p* = 0.762	F = 1.14; *p* = 0.286	F = 6.64; ***p*** **= 0.010**	F = 2.44; *p* = 0.119
	*FKBP5* × perceived self-efficacy × attachment	F = 2.14; *p* = 0.144	F = 3.17; *p* = 0.076	F = 0.06; *p* = 0.805	F = 1.57; *p* = 0.211	F = 6.18; ***p*** **= 0.013**	F = 2.18; *p* = 0.141
	R^2^	0.170	0.172	0.183	0.160	0.175	0.169

## Data Availability

Not applicable.

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
