# Peer review of "The Moderating Role of the FKBP5 Gene Polymorphisms in the Relationship between Attachment Style, Perceived Stress and Psychotic-like Experiences in Non-Clinical Young Adults"

_jcm, 2022, doi:10.3390/jcm11061614_

Round 1

Reviewer 1 Report

Being a very particular topic, I would try to better explain to the reader why these results can have such value in the scientific world.

Author Response

Thank you for efforts in reviewing our paper. According to your suggestions, we have added relevant paragraph in the conclusions subsection.

Yours sincerely,

Filip Stramecki

Reviewer 2 Report

This is a nicely presented study investigating the moderating role of the FKBP5 gene polymorphisms in the relationship between attachment style, perceived stress and psychotic-like experiences (PLEs) in non-clinical young Polish adults. Results showed that FKBP5 gene might moderate relationship between attachment, perceived stress and PLEs. Although the methods are relatively sound and the findings are clear, a flaw requires attention. In this study, all psychological tests, including the Psychosis Attachment Measure (PAM), the Perceived Stress Scale-10 (PSS-10) and the Prodromal Questionnaire 16 (PQ-16) were no Polish revision; therefore, authors should be presented in limitations and give explanations.

Author Response

Thank you for efforts in reviewing our paper. According to your suggestions, the following changes in the manuscript have been made:

  • We have highlighted more clearly that all questionnaires used in the present study were Polish versions.
  • We have performed additional language editing.

Yours sincerely,

Filip Stramecki

Reviewer 3 Report

Thank you for giving me the opportunity to review the manuscript.  I think that it is necessary to revise the manuscript based on the following points.

1) Please explain the design of the study, a cross-sectional study, in the method and abstract. Please clarify what is PECO and what is effect modifiers.

2) Please describe the relevant dates, including periods of recruitment, exposure, follow-up, and data collection.

3) Please describe any covariates, including predictors, potential confounders, and effect modifiers.

4) Why ANCOVA was selected as a method of statistical analysis in this study, although this study is an observational study?? 

5) Although there are so many covariates to be controlled for, I cannot find any statistical plan to control for covariates. Please explain it. If possible please use and analyze several models. For example, please investigate the association of an exposure and the outcome in the Model 1. please examine the association of them after controlling for baseline characteristics in Model 2. Please investigate the association of them after controlling for baseline characteristics and effect modifiers in Model 3.

5) The first sentence of "In last decades, numerous studies have focused on risk factors leading to psychosis development, and demonstrated the importance of [1-3]. " seemed grammatically incorrect. Please revise it.

6) In the Table 1, please explain the baseline characteristics of the Exposure group and the Control group. 

7) Although I am not a native speaker of English, English should be edited extensively.

This manuscript did not show any control group, any confounders, and what is the effect modifiers/mediators. Furthermore, it seemed that the authors did not analyzed them enough. Therefore, I think it is impossible to make any conclusion on the association among the FKBP5 gene polymorphisms, attachment style, perceived stress and psychotic-like experiences in the current form.

Author Response

Thank you for efforts in reviewing our paper. According to your suggestions, the following changes in the manuscript have been made:

  • We have explained more clearly the study design in the method and abstract subsections.
  • We added information about period of recruitment and highlighted the period of perceived stress assessed using the PSS-10.
  • Information about added covariates including age, gender, history of clinical diagnosis and frequent substance use are described in the statistical analysis subsection.
  • The ANCOVA was used due to expected effects of potential confounding factors, including age, gender, a history of clinical diagnosis and frequent substance use. This justification has been added to the manuscript.
  • Thank you for suggesting this point. We have outlined the plan of analyses in the statistics subsection. First, we tested main effect of exposure (perceived stress). Then, we added the effects of general characteristics as covariates (age, sex, clinical diagnosis and frequent substance use). Finally, we added the effects of moderators (attachment style, FKBP5 polymorphisms) as well as their two-way and three-way interactions (see the manuscript for details). 
  • The first sentence of the Introduction has been edited.
  • We have created additional Table.
  • We have performed additional language editing.

Yours sincerely,

Filip Stramecki

Round 2

Reviewer 3 Report

I think that this manuscript would be suitable for publication in this journal.